# Influence of Skin Marker Positioning and Their Combinations on Hip Joint Center Estimation Using the Functional Method

**DOI:** 10.3390/bioengineering11030297

**Published:** 2024-03-21

**Authors:** Lucas Martinez, Matthieu Lalevée, Thomas Poirier, Helena Brunel, Jean Matsoukis, Stéphane Van Driessche, Fabien Billuart

**Affiliations:** 1Unité de Recherche ERPHAN, UR 20201, UVSQ, 92380 Garches, France; f.billuart@ifmk.fr; 2Laboratoire d’Analyse du Mouvement, Institut de Formation en Masso-Kinésithérapie Saint Michel, 68 rue du Commerce, 75015 Paris, France; t.poirier@ifmk.fr (T.P.);; 3CETAPS UR3832, Research Center for Sports and Athletic Activities Transformations, University of Rouen Normandy, 76821 Mont-Saint-Aignan, France; matthieu.lalevee@gmail.com; 4Department of Orthopedic Surgery, Rouen University Hospital, 37 Bd Gambetta, 76000 Rouen, France; 5Département de Chirurgie Orthopédique, Groupe Hospitalier du Havre, BP24, 76083 Le Havre CEDEX, France; jean.matoukis@gmail.com; 6Polyclinique Sainte Marguerite, 5 Avenue de la Font Sainte-Marguerite, 89000 Auxerre, France; svandri@gmail.com; 7Université de Versailles-Saint-Quentin-en-Yvelines, UFR Simone Veil-Santé, 20 Avenue de la Source de la Bièvre, 78180 Montigny-le-Bretonneux, France

**Keywords:** gait analysis, skin marker positioning, hip joint center, functional method, orthopedics

## Abstract

Accurate estimation of hip joint center (HJC) position is crucial during gait analysis. HJC is obtained with predictive or functional methods. But in the functional method, there is no consensus on where to place the skin markers and which combination to use. The objective of this study was to analyze how different combinations of skin markers affect the estimation of HJC position relative to predictive methods. Forty-one healthy volunteers were included in this study; thirteen markers were placed on the pelvis and hip of each subject’s lower limbs. Various marker combinations were used to determine the HJC position based on ten calibration movement trials, captured by a motion capture system. The estimated HJC position for each combination was evaluated by focusing on the range and standard deviation of the mean norm values of HJC and the mean X, Y, Z coordinates of HJC for each limb. The combinations that produced the best estimates incorporated the markers on the pelvis and on proximal and easily identifiable muscles, with results close to predictive methods. The combination that excluded the markers on the pelvis was not robust in estimating the HJC position.

## 1. Introduction

Quantitative gait analysis (QGA) is recognized in various fields, such as total hip arthroplasty (THA), as the gold standard for evaluating postoperative gait perturbations [1,2]. THA is a well-accepted and successful procedure for patients suffering from hip osteoarthritis [3,4]. However, reconstruction of the hip modifies the hip’s geometry and these changes affect hip joint kinetics and kinematics. [5,6]. Thus, the location of the hip joint center (HJC) is crucial for quantifying musculoskeletal loading at the hip joint [3,7]. In fact, Stagni et al. [8] showed that errors in HJC positioning can affect hip kinematics. For gait analysis, two methods have historically been used to define HJC positioning [7,9,10,11,12,13,14,15,16].

The first and most widely used method [17,18] is the predictive method, which uses regression equations based on bone geometry of healthy subjects to derive the position of the HJC [7,11]. Conventional gait models such as the well-known Vicon Plug-in Gait (PiG) model use precise skin marker positioning combined with static calibration along with anthropometric data (length of lower limbs, distance between the two iliac spines) [11,13,19,20,21]. But this model has limitations. First, inaccurate skin marker positioning can distort the HJC positioning [22]. Also, changes in a person’s anthropometric data following surgery (e.g., THA) can induce inaccuracies when calculating the HJC (e.g., partial restoration of femoral offset and combined offset) [22].

The second method—the so-called functional method—uses a set of skin markers and dynamic calibration during precise, validated movements performed by a subject, such as the star-arc movement, for positioning the HJC [23]. This method is less dependent on accurate skin marker placement and does not require anthropometric data. However, its main limitation is soft tissue artefact (STA), which Hara et al. [24] defined as displacement of skin markers relative to the underlying bone landmarks due to muscle contraction, skin movement and inertia. The weighted optimal common shape technique (wOCST), developed for Vicon devices was shown to be a valid method for more accurately determining the joint centers by considering the non-uniform distribution of the STA [9,25,26]. OCST, first described by Taylor et al. in 2005 [27], is a statistical approach using a so-called Procrustes analysis to determine the best rigid marker configuration (having a stable shape during recording) that fits optimally to the marker positions over all time frames. The wOCST developed by Heller et al. [26] uses the standard OCST to suppress the STA and uses the Symmetrical Center of Rotation Estimation (SCoRE) for determining the HJC [9,25,26]. The SCoRE algorithm is used to determine the center of rotation of spherical joints and is based on the fact that a joint center is stationary within each segment. The SCoRE also applies a “weight” to each marker based on its contribution to HJC estimation. Therefore, these validated methods could be suitable for QGA in an orthopedic context such as THA as they do not require anthropometric measurements [3].

Although functional methods are recognized as being rapid, robust and precise, few studies provide information on the influence of skin marker positioning and their combinations on the estimation of HJC position [25]. Before functional methods can be incorporated into routine clinical use in the context of THA, we need to define how skin marker positioning and their combinations influence the estimation of HJC position and compare it to predictive method. This information can guide clinicians on the best skin marker positioning and combinations to use, while being compatible with the requirements of clinical follow-up.

For this study, starting from the optimal regions for marker placement defined by Kratzenstein et al. [25] to limit STA, eight different skin markers combinations were created. The objective of this study, carried out first in asymptomatic subjects, was to analyze how different combinations of skin markers affect the estimation of HJC position relative to predictive methods. We hypothesized that the skin marker combinations that produce the best estimate of HJC position compared to the predictive methods will be those that include (1) markers on the pelvic bone and (2) markers on easily identifiable muscles such as the anterior or/and posterior thigh muscles.

## 2. Materials and Methods

### 2.1. Subjects

The data were collected between January and May 2019. Forty-one asymptomatic subjects between 19 and 45 years of age were included (Table 1, Figure 1). All were physiotherapy students.

### 2.2. Methods

The inclusion criterion was a BMI < 30 kg/m^2^. The exclusion criteria were symptomatic orthopedic conditions in the lower limbs for less than 6 months, neurological or vestibular disease, visual, cardiovascular, respiratory, cognitive or psychiatric disorders, consumption of drugs or alcohol within the previous 24 h.

### 2.3. Measurement Protocol

#### 2.3.1. Anthropometric Data

The following anthropometric measurements were taken in each subject (in bare feet) to carry out the calibration for the predictive method, according to the Conventional Gait Model developed by Davis et al. [21]: height, weight, distance between AnteroSuperior Iliac Spine (ASIS) and medial malleolus (measured with a soft measuring tape in a standing subject after landmarks were palpated by an investigator with several years of continuous practice experience).

#### 2.3.2. Marker Placement

Thirteen 14 mm diameter “hard base type” reflective passive markers (Vicon Motion System Ltd., Oxford, UK) were stuck on each lower limb using hypoallergenic double-sided tape: 2 on the pelvis and 11 on the thigh (Figure 2). After anatomical palpation, the markers were applied by following the Vicon PiG guidelines [19,21,22] for the pelvis and the optimal regions described by Kratzenstein et al. for the thigh [25] (Figure 2). Table 2 describes the exact position of the markers on the pelvis and thigh. This marker positioning defined by Kratzenstein et al. [25] should limit STA.

#### 2.3.3. Data Acquisition

To obtain the coordinates (X, Y, Z) of the HJC, calibration trials were completed for each model:− Static calibration for the predictive methods.− Dynamic calibration using a star-arc movement [23] for the functional method.

For the static calibration, once the passive reflective markers had been attached, the subject was asked to stand with their feet hip-width apart and their arms held away from the body, to ensure that all the markers on the lower limbs were visible. For the dynamic calibration, each subject did a slow star-arc movement with each lower limb [3,23,28]. This movement was repeated and recorded ten times, thus a total of 20 movements for the two limbs. This calibration movement [3,23] corresponds to circumduction combining hip flexion/extension at 30° and abduction/adduction in an amplitude the subject could comfortably perform (minimum 15° and maximum 30° [23]). A slow movement speed as described by Begon et al. [28] was chosen for this study.

The two calibration tests were acquired using eight optical cameras (Bonita B10™, Vicon Motion System Ltd., Oxford, UK) and two videos cameras (Bonita 720C B10™, Vicon), and recorded simultaneously with Vicon Nexus™ software (version 2.5, Vicon) which incorporated the wOCST algorithms. The reflective skin markers were not removed and repositioned between each calibration trial.

#### 2.3.4. Data Processing

After having labelled and captured the trajectory of each skin marker, eight combinations of thigh and pelvis markers were defined, starting with the mandatory markers for PiG (ASIS, PSIS) [22] and the optimal regions for marker placement according to Kratzenstein et al. [25] (GTR, THIAP, THIAD, THILD, THIPP, THIPD, THI, KNE, PAT, MKNE). Each combination had a different number of markers and positioning as shown in Table 3 and Figure 2.

The HJC coordinates [19,21] were calculated based on a landmark whose center is midway between the left ASIS and right ASIS. The X axis (in red) passes through the center of the landmark and the middle of the left and right PSIS. The Y axis (green) passes through the middle of the landmark and right ASIS. The Z axis (blue) corresponds to the vectorial product between the X axis, Y axis and center of the left and right ASIS, oriented upwards.

#### 2.3.5. Estimate of the HJC by the Functional Method

In the Nexus™ software, the wOCST algorithm developed by Heller et al. [26] applies a weighting factor to each marker based on its contribution to representing the joint’s spherical movement. This approach incorporates the standard OCST described previously, which eliminates any movement of the markers relative to each other by generating a set of rigid markers starting from complete marker data for each combination and adding a SCoRE to it [9]. This determines the HJC based on movement data for two segments moving simultaneously.

Thus, for each of the ten calibration trials for each of the subjects’ limbs, eight HJC coordinates (X, Y, Z) were derived from the functional method.

#### 2.3.6. Estimate of the HJC by the Predictive Method

The HJC was also determined by predictive methods in two different manners: one via the linear regression method developed by Davis et al. [21], called “PiG” in this study and the other by the linear regression method developed by Harrington et al. [19], called “Harrington” in this study.

### 2.4. Statistical Analysis

Using the skin markers, eight combinations of markers were defined (Table 3, Figure 2) to calculate the X, Y, Z coordinates of their respective HJCs. Also, the HJC coordinates derived from the PiG and Harrington regression methods were calculated, for a total of ten HJCs.

Each subject did ten calibration trials for each limb. Thus, for each subject and each limb, the following were calculated based on these ten trials:− Mean norm (in mm) ± SD and range (max-min) for each combination.− Mean coordinates ± SD and range (max-min) in X, Y and Z of each combination.

Then, the following were calculated for each combination (on each lower limb):− Mean norms using the mean norms of each subject (in mm).− Mean SD of the norms using the mean SD of each subject.− Mean of the norm ranges using the mean ranges of each subject.− Mean coordinates (X, Y, Z) using the mean coordinates in X, Y, Z of each subject.− Mean SD of coordinates (X, Y, Z) using the mean SD in X, Y, Z of each subject.− Mean range of coordinates (X, Y, Z) using the mean range in X, Y, Z of each subject.

A Shapiro–Wilk test was performed to determine if the data were normally distributed. As normality was not guaranteed for certain variables, non-parametric tests were used. Kruskal–Wallis tests were used for comparisons among all combinations. Wilcoxon tests were used for pairwise comparisons between the combinations on the different parameters and each lower limb (R and L): norm, mean, SD and range of HJC positions in X, Y, Z. The significance threshold was set at α = 0.05. *p* values were corrected for multiple testing using the Bonferroni method. The corrected significance level was set to α = 0.001. The statistical analysis was performed using the software R™ (version 4.0.4, Bell Laboratories, Murray Hill, NY, USA).

## 3. Results

The entire dataset for this study is available on request from the corresponding author.

### 3.1. Comparison of Combinations to Each Other Based on Their Norms

Analyzing the mean norm of the various combinations eliminates the need to use the system of X, Y, Z axes and provides an initial view of the data. The mean of the norm of the C6 combination was significantly higher than that of the other combinations in the right and left limbs (except C1 and C2, Figure 3A,B, Table 4). C6 also had a mean range and SD that was significantly higher than the other combinations in the right and left limbs (except C1 and C3 for range and C4 on the right lower limb for SD, Figure 4A–D, Table 4).

### 3.2. Comparison between Combinations Based on Their Coordinates on the Different Axes

The second level of data analysis involves looking at the coordinates on the different axes. The mean SDs and ranges on the X axis were significantly higher than those on the Y and Z axes, no matter the combination and the limb considered (Figure 5 and Figure 6). In both limbs, on most axes (except SD and range on Y axis for the left lower limb), the C6 combination had significantly higher SDs and ranges relative to most of the other combinations (Figure 5 and Figure 6, Table 5 and Table 6).

### 3.3. Comparison with Predictive Models

The range and SD of the mean norm of the PiG and Harrington were significantly smaller than all the other combinations, whether the right or left limb was analyzed (Figure 4, Table 4). There were no significant differences between the two predictive methods for the mean of the norms, SDs and ranges (Table 4). There were also no significant differences between the two predictive methods for the SDs and ranges on X and Z axis (except for SD and range on Z axis for the right lower limb).

There were no significant differences between the two predictive methods and C5 and C8 for the mean of the norm. The mean SDs and ranges on all axes and both lower limbs were significantly higher for every combination except C5 and C8 than those derived from the two predictive methods (Table 5 and Table 6).

## 4. Discussion

Determining the HJC is a key focus of QGA particularly for a clinician assessing the postoperative consequences of THA. Of the methods available for determining the HJC, the so-called functional method requires specific skin marker placement and a calibration movement performed by the subject. While this method is well suited to routine clinical use, there is no evidence that it provides greater value than the so-called predictive methods. In this study, we chose to use methods for determining HJC that had been validated in the literature and were available in the gait analysis software owned by clinicians. Thus, the objective of this study was to guide the clinician using functional methods in the choice of skin markers and their combinations, in order to have an accurate estimate of the HJC position relative to predictive methods. Among the skin marker combinations created based on the study by Kratzenstein et al. [25], some use a large set of markers (C1 with 13 pelvis and thigh markers) while others use only bony markers (C2), which in principle, reduces their relevance for routine clinical use. Other combinations excluded bony markers (C6) while still others used different combinations of markers placed on prominent and easily palpable muscle tissue, along with the classic bone markers (C3, C4, C5, C7 and C8). To evaluate the estimate of HJC position, our analysis focused on the range and SD of the mean norms of each combination, followed by their mean coordinates, SD and range on each axis (X, Y, Z). The larger the SD and range for the different parameters, the less robust the estimate of the HJC position.

The first analysis compared the various combinations using the mean norm, which disregards the axes; it showed that the C6 combination was the least robust of all the combinations (Table 4). Also, its mean range and SD were greatly superior to the other combinations (Table 4). Based on this analysis, C6 should be eliminated from the possible combinations. Analyzing C6 on each axis confirms this conclusion. C6 does not use the pelvis bony markers, thus affecting the estimate of HJC position and confirming the usefulness of pelvis markers. These results validate the first part of our hypothesis, shared by most studies and seem logical [9,10,25,29]. Initially, this combination was proposed to overcome a clinician’s inability to palpate the bony landmarks of the pelvis in subjects with significant soft tissue thickness. For the clinician using functional methods in QGA, one of the key messages would be to keep these pelvic bone markers. However, it is interesting to note that these markers are not used during surgical navigation for knee arthroplasty. Even if marker tripods are fixed in the femur and tibia during this surgery—which reduces the measurement noise—given our findings, it seems important to keep the pelvis markers.

The C5 and C8 combinations were close to the estimate of the HJC position obtained by the predictive methods. These findings are a bit surprising given the literature. Indeed, Fiorentino et al. [11] thought that using skin markers on the proximolateral part of the thigh induced STA and modified the position of the HJC compared to the HJC obtained by fluoroscopy. However, their findings were predictable because the HJC obtained with fluoroscopy is more precise. It should be noted that this study did not use the same marker set, and a slightly different methodology to compute the HJC with functional method. C5 and C8 both use markers placed on large muscles such as the proximal and distal portions of the quadriceps (C5) and the proximal quadriceps and distal hamstrings (C8) (Table 2, THIAP, THIAD, THIPD), which, in theory, are highly susceptible to STA, therefore affecting the estimation of HJC position [9,23,30]. However, the use of wOCST and SCoRE seems to limit this phenomenon, as shown by Ehrig et al. and Heller et al. [9,26]. The proximal marker on the quadriceps, which is present in both combinations, may provide a specific clinical benefit, even though it should be particularly susceptible to STA according to previous publications [28]. These results seem to support the second part of our hypothesis on the relevance of skin markers on easily identifiable muscles such as anterior and/or posterior thigh muscles. The use of easily identifiable skin markers would thus facilitate clinical examination while achieving good precision during the estimation of the HJC position compared to predictive methods (Table 4, Table 5 and Table 6).

In addition, using all the markers (C1 combination) does not seem to improve the estimate of HJC position relative to combinations that use fewer markers such as C5 and C8 (Figure 3, Figure 4, Figure 5 and Figure 6). Kratzenstein et al. [25] similarly observed that increasing the number of markers does not improve the estimate of HJC position. This is relevant for clinicians: using a large number of markers extends the examination time without improving the estimate of HJC position. Likewise, the combination using only bone markers (C2) was not more robust in estimating the HJC position than those using soft tissue markers. This result is also relevant for clinicians because it shows that using only easily palpable bony markers does not improve the estimate of HJC position.

Even if this study provides interesting clinical elements, it has several limitations. First, we only evaluated the ability of different skin marker positions and their combinations to correctly estimate the position of the HJC compared to predictive methods. Since the skin markers were not removed and replaced between calibration trials, this study did not evaluate how easily the investigator could repeat the marker placement. This will be the topic of a future article. Second, the HJC positions, whether derived from the functional or predictive methods, were not compared with the true position of the HJC found with imaging (fluoroscopy or EOS^®^ for example). One can imagine that the HJC position obtained with functional methods will be more accurate in the context of arthroplasty as these methods are not impacted by alterations in the patient’s native hip (a patient who has undergone THA no longer meets the anthropometric standards used to determine the HJC with predictive methods). This will also be the topic of a future article since our study population consisted solely of young subjects who had no known pathologies. This is far removed from the context in which these combinations will be used. In fact, older subjects, who may present with muscle atrophy, excess weight, or restricted mobility secondary to hip arthritis or following THA surgery, may have difficulty performing the calibration movement. In the future, we plan to repeat this study with a larger number of subjects in an orthopedic context such as THA (pre- and postoperative), and to combine it with imaging (such as the EOS^®^ system). In this study, the sample is limited in size but it relatively large for a biomechanical modeling study compared to the literature [7,25,30]. Lastly, the speed of the calibration movement used here were not standardized. While there is broad agreement that the star-arc movement is the most repeatable, there is no information on which speed to use. Movements that are performed rapidly and/or with full range of motion appear to be less accurate when determining the HJC [28]. Similarly, Fiorentino et al. believe that a larger movement amplitude increases the artefacts [11,30]. Begon et al. [28] believe that higher speed increases STA. Nevertheless, in this study, the movements were performed slowly and not over the full range of motion.

## 5. Conclusions

It is very important to correctly estimate the HJC position, especially in a potential clinical follow-up setting such as hip arthroplasty. Using the functional method to estimate the HJC position eliminates the need for anthropometric data, which could be altered in an orthopedic setting. The objective of this study was to propose concrete solutions for orthopedic clinicians wishing to incorporate gait analysis in their clinical routine but who are limited by the pitfalls of predictive methods. The results of this study are clinically relevant—we have highlighted skin marker positions and combinations that alter the estimate of HJC position and others that improve it, while facilitating the clinical examination. The combination without pelvic bony markers should not be used because it does not provide a robust estimate of the HJC position, which confirms the first part of our hypothesis on the importance of these markers. Increasing the number of markers or using only bony markers does not improve the estimate of the HJC position compared to predictive methods. The other combinations using six markers per lower limb are preferred as they provide a good estimate of the HJC position compared to predictive methods, especially the two combinations (C5 and C8) that have a proximal marker on the rectus femoris. This element confirms the second part of our hypothesis on the relevance of placing skin markers on easily identifiable muscles such as anterior or/and posterior thigh muscles. Using a smaller number of markers in locations that are easy to palpate provides some initial proof of its potential use in routine clinical follow-up, although additional studies are needed with the target orthopedic population.

## Figures and Tables

**Figure 1 bioengineering-11-00297-f001:**
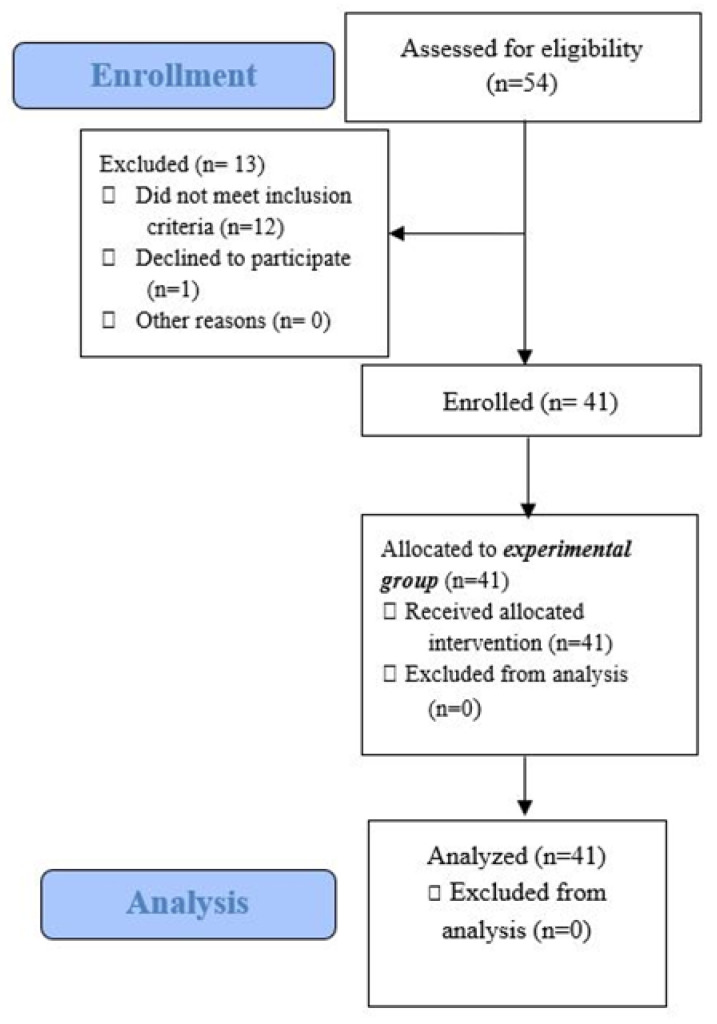
Study flow chart.

**Figure 2 bioengineering-11-00297-f002:**
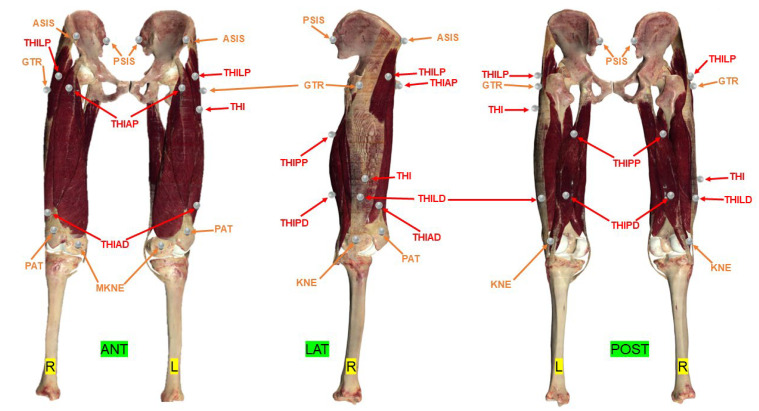
Location of passive reflective skin markers according to the PiG (by Davis et al. [21]) and Harrington models [19] of the pelvis and Kratzenstein et al. for the thigh [25]. The entire set of markers made up the combination C1. The marker name abbreviations are listed in Table 2. ANT = anterior view; LAT = lateral view; POST = posterior view; L = left; R= right. Red markers are muscular markers; Orange markers are bony markers.

**Figure 3 bioengineering-11-00297-f003:**
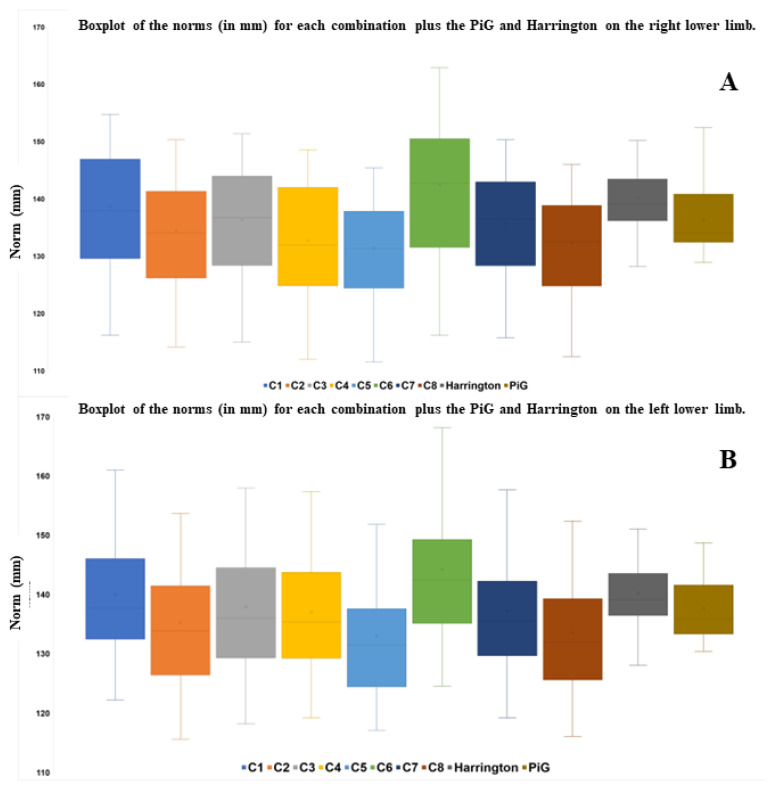
Boxplot of the norms (in mm) for each combination plus the PiG and Harrington on the right lower limb (**A**) and the left lower limb (**B**).

**Figure 4 bioengineering-11-00297-f004:**
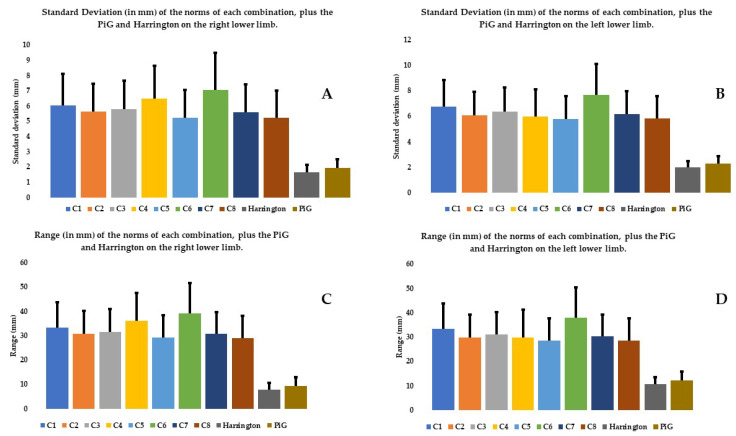
SD and range (in mm) on the right lower limb (**A**,**C**) and left (**B**,**D**) of the norms of each combination, plus the PiG and Harrington.

**Figure 5 bioengineering-11-00297-f005:**
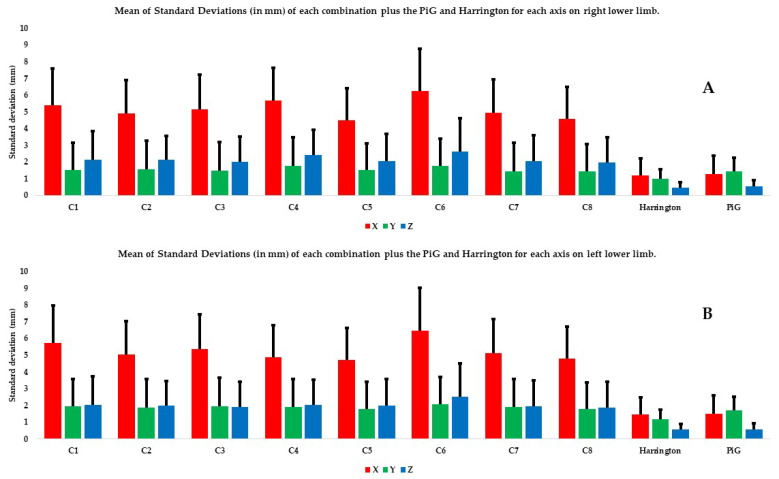
Mean of SDs (in mm) of each combination plus the PiG and Harrington for each axis and each lower limb: (**A**) the right lower limb and (**B**) the left lower limb. The X axis is in red, the Y axis in green, and the Z axis in blue.

**Figure 6 bioengineering-11-00297-f006:**
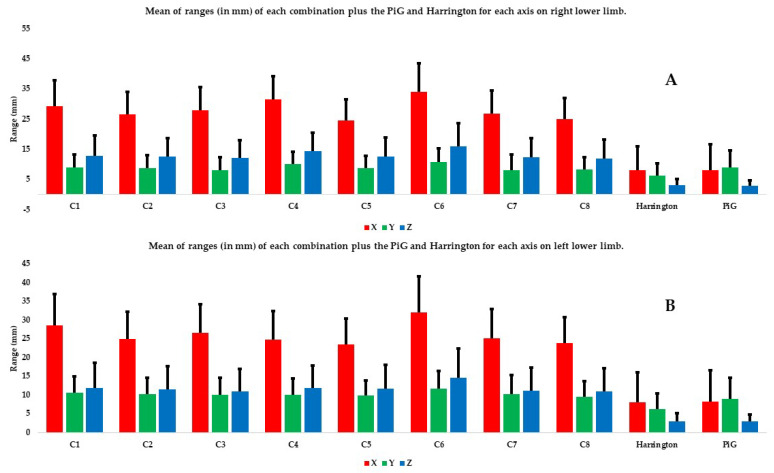
Mean of ranges (in mm) of each combination plus the PiG and Harrington for each axis and each lower limb: (**A**) the right lower limb and (**B**) the left lower limb. The X axis is in red, the Y axis in green, and the Z axis in blue.

**Table 1 bioengineering-11-00297-t001:** Characteristics of study group.

	Mean ± SD (min–max)
Number	41
Age (years)	**22.7 ± 5.69** (19–45)
Sex (Male (M)/Female (F))	20/21
Height (cm)	**172 ± 0.09** (168–182)
Mass (kg)	**64.80 ± 10.9** (59.45–83.56)
Body mass index (kg/m^2^)	**21.91 ± 2.76** (19.12–24.77)

(min–max) = minimum and maximum values of the parameter; SD: standard deviation.

**Table 2 bioengineering-11-00297-t002:** Positioning of skin markers on the pelvis and thigh.

Segment	Number of Markers	Marker Name	Anatomical Landmark
*Pelvis*	2 (4 on both sides)	ASIS	Anterosuperior iliac spine [21]
		PSIS	Posterosuperior iliac spine [21]
*Thigh*	11 (22 on both sides)	GTR	Greater trochanter [21]
		THIAP	Proximally from the belly of the rectus femoris = ***AI*** in Kratzenstein et al. [25]
		THIAD	Anterolateral area of the distal thigh = ***AII*** in Kratzenstein et al. [25]
		THILP	Proximally along the tensor fascia lata = ***LI*** in Kratzenstein et al. [25]
		THILD	Distally along the tensor fascia lata = ***LII*** in Kratzenstein et al. [25]
		THIPP	Proximal to the belly of the biceps femoris and semitendinosus = ***PI*** in Kratzenstein et al. [25]
		THIPD	Distal to the belly of the biceps femoris and semitendinosus = ***PII*** in Kratzenstein et al. [25]
		THI	Right side: inferior third of lateral portion of thigh Left side: superior third of lateral portion of thigh
		KNE	Lateral femoral condyle [21]
		PAT	Middle of superior edge of patella [21]
		MKNE	Medial femoral condyle [21]

**Table 3 bioengineering-11-00297-t003:** Combinations created from the various pelvis and thigh markers.

Name of Combination	Markers Included	Potential Clinical Benefit
**C1** = all the markers	All the markers	Will using all the markers improve the accuracy of the HJC estimate compared to predictive methods?
**C2** = markers on the bony landmarks	GTR MKNE KNE + pelvis	Simplify method by using only bone markers (easily palpable)
**C3** = proximal and distal posterior markers	GTR MKNE KNE + pelvis THIPP THIPD	Simplify method by using only the lateral, anterior, or posterior skin markers
**C4** = lateral markers	GTR MKNE KNE + pelvis THILP THILD	
**C5** = proximal and distal anterior markers	GTR MKNE KNE + pelvis THIAP THIAD	
**C6** = proximal and distal anterior markers without the pelvis markers	GTR MKNE KNE THIAP THIAD	Potentially useful if the pelvic bony landmarks cannot be palpated (too much soft tissue on the bony landmarks)
**C7** = anterior distal and posterior proximal markers	GTR MKNE KNE + pelvis THIAD THIPP	The THIAP, THIAD, THIPP, and THIPD markers are located in the corresponding regions on the muscle bellies (anterior: quadriceps; posterior: hamstrings). These locations are easier to palpate but could be affected by STA
**C8** = anterior proximal and posterior distal markers	GTR MKNE KNE + pelvis THIAP THIPD	

**Table 4 bioengineering-11-00297-t004:** *p* values for the pairwise comparisons of the range, mean and SD of the norm for each combination in the left and right lower limbs.

***p* value range of the norm for each combination in the right lower limb**											***p* value range of the norm for each combination in the left lower limb**										
	**C1**	**C2**	**C3**	**C4**	**C5**	**C6**	**C7**	**C8**	**H**	**P**		**C1**	**C2**	**C3**	**C4**	**C5**	**C6**	**C7**	**C8**	**H**	**P**
**C1**											**C1**										
**C2**											**C2**										
**C3**											**C3**										
**C4**											**C4**										
**C5**											**C5**										
**C6**											**C6**										
**C7**											**C7**										
**C8**											**C8**										
**H**											**H**										
**P**											**P**										
***p* value mean of the norm for each combination in the right lower limb**											***p* value mean of the norm for each combination in the left lower limb**										
	**C1**	**C2**	**C3**	**C4**	**C5**	**C6**	**C7**	**C8**	**H**	**P**		**C1**	**C2**	**C3**	**C4**	**C5**	**C6**	**C7**	**C8**	**H**	**P**
**C1**											**C1**										
**C2**											**C2**										
**C3**											**C3**										
**C4**											**C4**										
**C5**											**C5**										
**C6**											**C6**										
**C7**											**C7**										
**C8**											**C8**										
**H**											**H**										
**P**											**P**										
***p* value SD of the norm for each combination in the right lower limb**											***p* value SD of the norm for each combination in the left lower limb**										
	**C1**	**C2**	**C3**	**C4**	**C5**	**C6**	**C7**	**C8**	**H**	**PiG**		**C1**	**C2**	**C3**	**C4**	**C5**	**C6**	**C7**	**C8**	**H**	**P**
**C1**											**C1**										
**C2**											**C2**										
**C3**											**C3**										
**C4**											**C4**										
**C5**											**C5**										
**C6**											**C6**										
**C7**											**C7**										
**C8**											**C8**										
**H**											**H**										
**P**											**P**										
Red: *p* > 0.001; Green = *p* < 0.001. *p* = Plug-in Gait; H = Harrington																					

**Table 5 bioengineering-11-00297-t005:** *p* values for the pairwise comparisons of the mean SD of the X, Y, and Z coordinates in the left and right lower limbs.

***p* value SD of X coordinates in the right lower limb**											***p* value SD of X coordinates in the left lower limb**										
	**C1**	**C2**	**C3**	**C4**	**C5**	**C6**	**C7**	**C8**	**H**	**P**		**C1**	**C2**	**C3**	**C4**	**C5**	**C6**	**C7**	**C8**	**H**	**P**
**C1**											**C1**										
**C2**											**C2**										
**C3**											**C3**										
**C4**											**C4**										
**C5**											**C5**										
**C6**											**C6**										
**C7**											**C7**										
**C8**											**C8**										
**H**											**H**										
**P**											**P**										
***p* value SD of Y coordinates in the right lower limb**											***p* value SD of Y coordinates in the left lower limb**										
	**C1**	**C2**	**C3**	**C4**	**C5**	**C6**	**C7**	**C8**	**H**	**P**		**C1**	**C2**	**C3**	**C4**	**C5**	**C6**	**C7**	**C8**	**H**	**P**
**C1**											**C1**										
**C2**											**C2**										
**C3**											**C3**										
**C4**											**C4**										
**C5**											**C5**										
**C6**											**C6**										
**C7**											**C7**										
**C8**											**C8**										
**H**											**H**										
**P**											**P**										
***p* value SD of Z coordinates in the right lower limb**											***p* value SD of Z coordinates in the left lower limb**										
	**C1**	**C2**	**C3**	**C4**	**C5**	**C6**	**C7**	**C8**	**H**	**P**		**C1**	**C2**	**C3**	**C4**	**C5**	**C6**	**C7**	**C8**	**H**	**P**
**C1**											**C1**										
**C2**											**C2**										
**C3**											**C3**										
**C4**											**C4**										
**C5**											**C5**										
**C6**											**C6**										
**C7**											**C7**										
**C8**											**C8**										
**H**											**H**										
**P**											**P**										
Red: *p* > 0.001; Green = *p* < 0.001. *p* = Plug-in Gait; H = Harrington																					

**Table 6 bioengineering-11-00297-t006:** *p* values for the pairwise comparisons of the mean range of the X, Y, and Z coordinates in the left and right lower limbs.

***p* value range of X coordinates in the right lower limb**											***p* value range of X coordinates in the left lower limb**										
	**C1**	**C2**	**C3**	**C4**	**C5**	**C6**	**C7**	**C8**	**H**	**P**		**C1**	**C2**	**C3**	**C4**	**C5**	**C6**	**C7**	**C8**	**H**	**P**
**C1**											**C1**										
**C2**											**C2**										
**C3**											**C3**										
**C4**											**C4**										
**C5**											**C5**										
**C6**											**C6**										
**C7**											**C7**										
**C8**											**C8**										
**H**											**H**										
**P**											**P**										
***p* value range of Y coordinates in the right lower limb**											***p* value range of Y coordinates in the left lower limb**										
	**C1**	**C2**	**C3**	**C4**	**C5**	**C6**	**C7**	**C8**	**H**	**P**		**C1**	**C2**	**C3**	**C4**	**C5**	**C6**	**C7**	**C8**	**H**	**P**
**C1**											**C1**										
**C2**											**C2**										
**C3**											**C3**										
**C4**											**C4**										
**C5**											**C5**										
**C6**											**C6**										
**C7**											**C7**										
**C8**											**C8**										
**H**											**H**										
**P**	NA	NA	NA	NA	NA	NA	NA	NA	NA	NA	**P**	NA	NA	NA	NA	NA	NA	NA	NA	NA	
***p* value range of Z coordinates in the right lower limb**											***p* value range of Z coordinates in the left lower limb**										
	**C1**	**C2**	**C3**	**C4**	**C5**	**C6**	**C7**	**C8**	**H**	**P**		**C1**	**C2**	**C3**	**C4**	**C5**	**C6**	**C7**	**C8**	**H**	**P**
**C1**											**C1**										
**C2**											**C2**										
**C3**											**C3**										
**C4**											**C4**										
**C5**											**C5**										
**C6**											**C6**										
**C7**											**C7**										
**C8**											**C8**										
**H**											**H**										
**P**											**P**										
Red: *p* > 0.001; Green = *p* < 0.001. *p* = Plug-in Gait; H = Harrington																					

## Data Availability

The data presented in this study are available on request from the corresponding author.

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
