# Peer review of "Influence of Skin Marker Positioning and Their Combinations on Hip Joint Center Estimation Using the Functional Method"

_bioengineering, 2024, doi:10.3390/bioengineering11030297_

Round 1

Reviewer 1 Report

Comments and Suggestions for Authors

Point 1:- Keep Vertical error bar in positive bar only in all bar graph figures. (figure 4(c,d,e,f) Figure 5 and 6) 

Point 2:- In line 275-278, why the sentences were written in italic font , is it any special sentences to be highlighted 

Point 3:- in e C5 and C8 combinations you had got closed to the estimation of the HJC positioning but you mention it was in inconsistent with previous publi- 304 cations, So what makes your combination good one.

Author Response

Thank you for this thorough reading, which will greatly improve the quality of the manuscript. Below, you will find our responses to all of your comments and our subsequent modifications. We hope that this new version will meet your expectations. We are at your disposal. We have highlighted the corrections in yellow in the text. Please see attachment

Point 1:- Keep Vertical error bar in positive bar only in all bar graph figures. (figure 4(c,d,e,f) Figure 5 and 6) 

We have modified the figures in accordance with your comments. Please see Figures 4, 5 and 6.

Point 2:- In line 275-278, why the sentences were written in italic font , is it any special sentences to be highlighted 

Thank you for this remark. This is indeed an important sentence for the results, but it is not necessary to have it in italic font. We put it back into regular font.

Please see lines 264-265: “There were no significant differences between the two predictive methods and C5 and C8 for the mean of the norm

Point 3: - in e C5 and C8 combinations you had got closed to the estimation of the HJC positioning but you mention it was in inconsistent with previous publications, So what makes your combination good one.

Thank you for this very interesting remark. Indeed, combinations C5 and C8 are the closest to predictive methods. Looking at the literature, Fiorentino et al. showed that functional methods were less precise than fluoroscopy for estimating the HJC. This result was predictable in the sense that fluoroscopy using imaging will necessarily be more precise. However, the authors of this study did not compare the functional and predictive methods and did not use the same marker set or the wOCST. The term “inconsistent” that we used is misused and we have changed the sentence (Please see lines : 313-318: These findings are a bit surprising given the literature. Indeed, Fiorentino et al. [11] thought that using skin markers on the proximolateral part of the thigh induced STA and modified the position of the HJC compared to the HJC obtained by fluoroscopy. However, their findings were predictable because the HJC obtained with fluoroscopy is more precise. It should be noted that this study did not use the same marker set, and a slightly different methodology to compute the HJC with functional method”).

We believe that what makes our combinations good ones is using both easily identifiable bone markers and muscle skin markers and removing the STA with wOCST and SCoRE and above all, being as accurate as predictive methods. However, we were not able to test our combinations with fluoroscopy or other imaging techniques. This will be the topic of our next study.

Reviewer 2 Report

Comments and Suggestions for Authors

 Thank you for submitting your research paper titled "Influence of skin markers positioning and their combinations on the estimation of the Hip Joint Center with the functional method." We appreciate the effort and contribution you have made in exploring new treatment options. Upon reviewing your manuscript, we find that the overall content and methodology of the study are commendable. 

Compare with the Related article as:

doi: 10.34172/ijbsm.2021.08

 What specific skin markers were used in the study to estimate the Hip Joint Center (HJC)?

How were the skin markers positioned on the subjects' bodies, and what criteria were considered in their placement?

Can you explain the functional method used to estimate the HJC and how it was affected by the positioning of the skin markers?

Were different combinations of skin markers evaluated in the study? If so, what were the different combinations, and how did they influence the accuracy of the HJC estimation?

Comments on the Quality of English Language

 Thank you for submitting your research paper titled "Influence of skin markers positioning and their combinations on the estimation of the Hip Joint Center with the functional method." We appreciate the effort and contribution you have made in exploring new treatment options. Upon reviewing your manuscript, we find that the overall content and methodology of the study are commendable. 

Compare with the Related article as:

doi: 10.34172/ijbsm.2021.08

 What specific skin markers were used in the study to estimate the Hip Joint Center (HJC)?

How were the skin markers positioned on the subjects' bodies, and what criteria were considered in their placement?

Can you explain the functional method used to estimate the HJC and how it was affected by the positioning of the skin markers?

Were different combinations of skin markers evaluated in the study? If so, what were the different combinations, and how did they influence the accuracy of the HJC estimation?

Author Response

Thank you for this thorough reading, which will greatly improve the quality of the manuscript. Below, you will find our responses to all of your comments and our subsequent modifications. You made comments about the quality of the English language. We asked our English-language expert to review the revised manuscript before re-submission. Please see attachment

Compare with the Related article as:

doi: 10.34172/ijbsm.2021.08

The DOI number you provided refers us to this article: “A Review of Polymeric Wound Dress for the Treatment of Burns and Diabetic Wounds”. Is that correct?

What specific skin markers were used in the study to estimate the Hip Joint Center (HJC)?

We used 14-mm diameter “hard base type” reflective passive markers (Vicon Motion System Ltd, Oxford, UK). They were stuck using hypoallergenic double-sided tape.

We modified the article (please see lines: 135-137: “Thirteen 14-mm diameter “hard base type” reflective passive markers (Vicon Motion System Ltd, Oxford, UK) were stuck on each lower limb using hypoallergenic double-sided tape”)

How were the skin markers positioned on the subjects' bodies, and what criteria were considered in their placement?

The skin markers were placed as follows: after anatomical palpation, markers were placed according to the guidelines (provided by the manufacturers) by a clinician with several years of continuous practice experience. For the criteria: all methods carried out in this study were in accordance with the guidelines for gait analysis in clinical practice developed by Davis et al. We modified the text (please see lines : 131-138: “measured with a soft measuring tape in a standing subject after landmarks were palpated by an investigator with several years of continuous practice experience") Marker placement : Thirteen 14-mm diameter “hard base type” reflective passive markers (Vicon Motion System Ltd, Oxford, UK) were stuck on each lower limb using hypoallergenic double-sided tape: 2 on the pelvis and 11 on the thigh (Figure 2). After anatomical palpation, the markers were applied by following the Vicon PiG guidelines”)

Can you explain the functional method used to estimate the HJC and how it was affected by the positioning of the skin markers?

With functional methods (that we used in this study), the relative motion between two segments is the basis for a mathematically derived optimal location of the HJC. These methods assume the hip to be a ball joint.  This method requires less precision in the placement of skin markers and does not require anthropometric data, which is important in a clinical setting. But there is no consensus in the literature on the placement of skin markers and its main limitation is STA. Nevertheless, the Optimal Common Shape Technique (OCST) that we used in this study has been demonstrated as a reliable approach to improve the determination of the HJC in vivo. This technique (module available in Nexus software) considers a non-uniform distribution of STA and reduces them using OCST. Positioning markers on the thigh is a major risk of STA. However, we believed that OCST would be able to correct STA due to thigh markers. OCST is first described by Taylor et al. in 2005 [26] in animals, is a statistical approach using a so-called Procrustes analysis to determine an optimal rigid marker configuration (having a stable shape during recording) that fits optimally to the marker positions over all time frames. We completed the OCST by determining the SCoRE (Symmetrical Center of Rotation Estimation), an assessment of the accuracy of HJC estimation. The SCoRE is an algorithm (also available in nexus software) to determine the center of rotation of spherical joints and is based on the fact that a joint center is stationary in the reference (parent) segment. The SCoRE also applies a “weight” to each marker based on its contribution to the HJC estimation. It thus makes it possible to evaluate the capacity of the OCST to reduce the STA and thus determine the HJC in our case. The use of the SCoRE has been demonstrated to be the most accurate technique when both segments move simultaneously [9]. We made the hypothesis that the functional method is a reliable method for determining HJC. For the positioning of our markers via the functional method we used the recommendations developed by Kratzenstein et al. [25]. These authors define “ideal placement zones” on the thigh, where STA would be limited. We wanted to obtain the best compromise between simple marker placement (on large superficial muscles associated with bone markers) and HJC estimation as close as the predictive methods. Our hypothesis seems to be confirmed.

We modified the article. Please see lines 71-80, 177:

Lines 71-80: “OCST, first described by Taylor et al. in 2005 [27], is a statistical approach using a so-called Procrustes analysis to determine the best rigid marker configuration (having a stable shape during recording) that fits optimally to the marker positions over all time frames. The wOCST developed by Heller et al. [26] uses the standard OCST to suppress the STA and uses the Symmetrical Center of Rotation Estimation (SCoRE) for determining the HJC [9,25,26]. The SCoRE algorithm is used to determine the center of rotation of spherical joints and is based on the fact that a joint center is stationary within each segment. The SCoRE also applies a “weight” to each marker based on its contribution to HJC estima-tion. Therefore, these validated methods could be suitable for QGA in an orthopedic con-text such as THA as they do not require anthropometric measurements”

Line 177: “This approach incorporates the standard OCST described previously”

Were different combinations of skin markers evaluated in the study? If so, what were the different combinations, and how did they influence the accuracy of the HJC estimation?

Indeed, we evaluated several sets of markers in this study. We first placed our skin markers on each subject using the pelvic, femur and patella bone markers proposed by the Vicon PiG module and then we positioned markers according to the “ideal” placement zones defined by Kratzenstein et al. [25] We have given the details of the placement of the skin markers in Table 2 and Figure 2. Then we selected different combinations of markers to calculate the HJC. The combinations of markers and their potential value are presented in Table 3. Regarding their influence on the accuracy of the HJC estimate, we concluded that selecting all the markers (combination C1) has no benefit because it did not improve the accuracy of the HJC estimate compared to combinations using fewer markers. For clinicians this is important because the use of a high number of markers extends the examination time without increasing the precision of the measurements. The C2 combination only used the bony markers. In theory, this combination should be useful for clinicians because they are easily palpable. However, our study showed that they did not improve the accuracy of the HJC estimate, therefore of no value. With the C3, C4 and C5 combinations we proposed to simplify the protocol by only taking muscular markers on one side of the thigh in addition to the bony markers. This could have helped the clinician in positioning the markers. C3 and C4 did not show superiority over C5 and C8 or between them. They are therefore of no value. The C6 combination aimed to not exclude the pelvic markers if they were not palpable. It turns out that it profoundly affects the estimate of the HJC, it should therefore be eliminated. C7 and C8 preserved bony markers and added easily identifiable muscle markers on the lower limb. We assumed that they would facilitate the clinical examination. Of the two combinations, only C8 was as accurate as predictive methods for estimating the HJC.  Finally, only C5 and C8 are as precise as the predictive methods for estimating the position of the HJC.

We added these elements to the manuscript (please see lines: 313-318, 334-339, 371-380)

Lines 313-318: “ These findings are a bit surprising given the literature. Indeed, Fiorentino et al. [11] thought that using skin markers on the proximolateral part of the thigh induced STA and modified the position of the HJC compared to the HJC obtained by fluoroscopy. However, their findings were predictable because the HJC obtained with fluoroscopy is more precise. It should be noted that this study did not use the same marker set, and a slightly different methodology to compute the HJC with functional method.”

Lines 334-339: “This is relevant for clinicians: using a large number of markers extends the examination time without improving the estimate of HJC position. Likewise, the combination using only bone markers (C2) was not more robust in estimating the HJC position than those using soft tissue markers. This result is also relevant for clinicians because it shows that using only easily palpable bony markers does not improve the estimate of HJC position.”

Lines 371-380: “The objective of this study was to propose concrete solutions for orthopedic clinicians wishing to incorporate gait analysis in their clinical routine but who are limited by the pitfalls of predictive methods. The results of this study are clinically relevant—we have highlighted skin marker positions and combinations that alter the estimate of HJC posi-tion and others that improve it, while facilitating the clinical examination. The combina-tion without pelvic bony markers should not be used because it does not provide a robust estimate of the HJC position, which confirms the first part of our hypothesis on the im-portance of these markers. Increasing the number of markers or using only bony markers does not improve the estimate of the HJC position compared to predictive methods”

Reviewer 3 Report

Comments and Suggestions for Authors

These are the comments for the paper.

1. The authors need to address the value of this research where the methods employed a  small number of subjects in the study.

2.  How were the methods in this study validated  and compared to current models or approaches ?

3.  I suggest to use other mathematical methods to increase the novelty of this paper.

Comments on the Quality of English Language

none

Author Response

Thank you for this thorough reading, which will greatly improve the quality of the manuscript. Below, you will find our responses to all of your comments and our subsequent modifications. You made comments about the quality of the English language. We asked our English-language expert to review the revised manuscript before re-submission. Please see attachment

These are the comments for the paper.

  1. The authors need to address the value of this research where the methods employed a small number of subjects in the study.

Thank you for this remark. This study is preliminary to a series of future studies that will follow. The objective of this project is to propose concrete solutions to orthopedic clinicians wishing to use gait analysis in their clinical routine but who are limited by the pitfall of predictive methods mentioned in the introduction. We want to define marker positions and combinations that can be used routinely in a clinical setting (reduced number of markers and easier placement) while maintaining a high level of precision for estimating the HJC. In this sense this work is innovative because in this specific context there is no comparable work in the literature.

We added this point in the introduction and discussion Please see line: 313-318, 328-330, 334-339, 371-380

Lines 313-318: “ These findings are a bit surprising given the literature. Indeed, Fiorentino et al. [11] thought that using skin markers on the proximolateral part of the thigh induced STA and modified the position of the HJC compared to the HJC obtained by fluoroscopy. However, their findings were predictable because the HJC obtained with fluoroscopy is more precise. It should be noted that this study did not use the same marker set, and a slightly different methodology to compute the HJC with functional method.”

Lines 328-330: “The use of easily identifiable skin markers would thus facilitate clinical examination while achieving good precision during the estimation of the HJC position compared to predictive methods (Tables 4, 5, 6).”

Lines 334-339: “This is relevant for clinicians: using a large number of markers extends the examination time without improving the estimate of HJC position. Likewise, the combination using only bone markers (C2) was not more robust in estimating the HJC position than those using soft tissue markers. This result is also relevant for clinicians because it shows that using only easily palpable bony markers does not improve the estimate of HJC position.”

Lines 371-380: “The objective of this study was to propose concrete solutions for orthopedic clinicians wishing to incorporate gait analysis in their clinical routine but who are limited by the pitfalls of predictive methods. The results of this study are clinically relevant—we have highlighted skin marker positions and combinations that alter the estimate of HJC posi-tion and others that improve it, while facilitating the clinical examination. The combina-tion without pelvic bony markers should not be used because it does not provide a robust estimate of the HJC position, which confirms the first part of our hypothesis on the im-portance of these markers. Increasing the number of markers or using only bony markers does not improve the estimate of the HJC position compared to predictive methods”

Indeed, our sample is limited in size, however it remains high for a biomechanical modeling study. Most of the seminal works in this field present smaller samples compared to our study: Fiorentino et al. used 11 subjects in their study, Kratzenstein et al. used 7 subjects, and Sangeux et al. used 17 subjects. Obviously, we would like to have a larger sample for our future studies.

We added this point in the discussion, please see lines 358-359: “In this study, the sample is limited in size but it relatively large for a biomechanical mod-eling study compared to the literature [7,25,30].”

  1. How were the methods in this study validated  and compared to current models or approaches ?

For this study, we did not create new methods for computing HJC. We used methods validated in the literature (Heller et al. for the wOCST, Kratzenstein et al. for marker positioning, Davis et al. and Harrington et al. for predictive methods), which do not require a validation step for this study. Marker positioning also existed in the literature. The innovation lies in the different combinations proposed to estimate the HJC position. We made this choice to compare our data with the existing literature. The comparison is only made with predictive methods.

We have added literature references in the Materials & Methods: lines 177, 180, Table 2.

Line 177: “This approach incorporates the standard OCST described previously”

Line 180: reference [9]

  1. I suggest to use other mathematical methods to increase the novelty of this paper.

This point is indeed interesting, and we thank you for this remark. We did not want to develop new mathematical methods in this study because our goal was to remain accessible to clinicians using Gait Analysis. We wanted to use validated mathematical methods that are, above all, available in the software offered by manufacturers such as Vicon Nexus for example. We plan to develop new methods in future studies. We added this item to the discussion. Please see lines: 283-284: “In this study, we chose to use methods for determining HJC that had been validated in the literature and were available in the gait analysis software owned by clinicians.”

Round 2

Reviewer 3 Report

Comments and Suggestions for Authors

The authors have addressed the missing gaps.